# Tunable Visible Light and Energy Transfer Mechanism in Tm^3+^ and Silver Nanoclusters within Co-Doped GeO_2_-PbO Glasses

**DOI:** 10.3390/mi14112078

**Published:** 2023-11-09

**Authors:** Marcos Vinicius de Morais Nishimura, Augusto Anselmo Amaro, Camila Dias da Silva Bordon, Jessica Dipold, Niklaus Ursus Wetter, Luciana Reyes Pires Kassab

**Affiliations:** 1Departamento de Engenharia de Sistemas Eletrônicos, Escola Politécnica da Universidade de São Paulo, Av. Prof. Luciano Gualberto, 158, Travessa 3, São Paulo 05508-900, SP, Brazilcamiladsb@usp.br (C.D.d.S.B.); 2Instituto de Pesquisas Energéticas e Nucleares, IPEN-CNEN, 2242, Av. Prof. Lineu Prestes, São Paulo 05508-000, SP, Brazil; jessica.dipold@alumni.usp.br (J.D.); nuwetter@ipen.br (N.U.W.); 3Faculdade de Tecnologia de São Paulo, CEETEPS, Praça Cel. Fernando Prestes, 30, São Paulo 01124-060, SP, Brazil; kassablm@fatecsp.br

**Keywords:** germanate glasses, silver nanoclusters, rare-earth ions, Thulium ions, tunable luminescence, melt-quenching technique

## Abstract

This study introduces a novel method for producing Ag nanoclusters (NCs) within GeO_2_-PbO glasses doped with Tm^3+^ ions. Sample preparation involved the melt-quenching method, employing adequate heat treatment to facilitate Ag NC formation. Absorption spectroscopy confirmed trivalent rare-earth ion incorporation. Ag NC identification and the amorphous structure were observed using transmission electron microscopy. A tunable visible emission from blue to the yellow region was observed. The energy transfer mechanism from Ag NCs to Tm^3+^ ions was demonstrated by enhanced 800 nm emission under 380 and 400 nm excitations, mainly for samples with a higher concentration of Ag NCs; moreover, the long lifetime decrease of Ag NCs at 600 nm (excited at 380 and 400 nm) and the lifetime increase of Tm^3+^ ions at 800 nm (excitation of 405 nm) corroborated the energy transfer between those species. Therefore, we attribute this energy transfer mechanism to the decay processes from S_1_→T_1_ and T_1_→S_0_ levels of Ag NCs to the ^3^H_4_ level of Tm^3+^ ions serving as the primary path of energy transfer in this system. GeO_2_-PbO glasses demonstrated potential as materials to host Ag NCs with applications for photonics as solar cell coatings, wideband light sources, and continuous-wave tunable lasers in the visible spectrum, among others.

## 1. Introduction

Glasses play a pivotal role in photonic device development and optical components due to their ease of manipulation and their ability to take on various shapes and sizes. Additionally, glasses offer unique characteristics, including exceptional transparency and the convenience of doping with active ions. Within this context, silver-containing amorphous materials have emerged as promising candidates for light-emitting devices. Silver (Ag) nanoclusters (NCs) embedded within glass matrices have demonstrated significant potential for photonic applications, including both undoped and rare-earth (RE)-doped configurations, all produced using the melt-quenching technique [1].

Initially, research efforts primarily focused on investigating Ag NCs within liquid, polymer, or organic matrices [2,3]. However, recent developments have shifted attention towards oxyfluoride glasses as the preferred host materials for Ag NCs [4,5,6,7,8,9,10,11,12]. This shift is primarily attributed to the advantageous properties of oxyfluoride glasses, particularly for fibers, thin films, and various manufacturing techniques with respect to alternative materials [5]. Consequently, the scientific literature has witnessed a surge in exploration involving diverse glass compositions as potential hosts for Ag NCs. These compositions span zinc borate CABAl [13], fluorophosphate [14,15,16,17,18,19], borates [20], and borosilicates [21,22], among other glass matrices [23,24,25,26,27], thereby broadening the possibilities of Ag NC-based photonic applications. The applications of this technology span a wide array of photonic advancements, including photovoltaic devices, flexible screens, sources for white light [28], fiber [29,30], including wavelength tuning [31,32]. It is important to note that Ag NCs stand apart from metallic nanoparticles (NPs) due to their unique composition, consisting of only small agglomerates of silver atoms, resulting in an amorphous structural arrangement [16]. This distinctiveness sets them apart from metallic NPs, which are characterized by their crystalline nature, larger physical dimensions, and the presence of an absorption plasmon band [33,34]. The discrete energy level of Ag NCs contributes to their exceptional photoluminescent (PL) properties, which are sensitive to various factors, including the excitation wavelength, glass composition, Ag dopant concentration, and the localized density of Ag NCs. It is worth noting that higher concentrations of Ag NCs can lead to agglomeration, potentially culminating in the formation of metallic NPs that can result in the coexistence of NCs and NPs simultaneously in the material.

Ag NCs exhibit distinctive optical properties that differentiate them from bulk silver and individual silver atoms. While bulk silver typically does not display luminescence, isolated silver atoms generate PL bands that normally cover the UV-blue region. In contrast, Ag NCs characteristically produce a broad PL band that encompasses the visible portion of the electromagnetic spectrum [2,7]. Furthermore, the PL characteristics of Ag NCs are intricately tied to their size and provide tuning across the UV to visible and near-infrared (NIR) regions by carefully selecting the appropriate excitation wavelength [31]. Both Ag NCs and Ag NPs have shown the ability to enhance the PL intensity of lanthanide ions. In the case of Ag NPs, this enhancement is due to the generation of an amplified local electromagnetic field in the neighborhood of the RE ions [35,36]. Conversely, in the case of Ag NCs, mechanisms involving direct energy transfer (ET) [9,10,11,12,13,18,19,20,21,22,27] and Förster resonance ET [15] have been reported as the means of achieving this enhancement. In certain instances, the overlap of the PL bands related to Ag NCs and the ones corresponding to RE ions can result in the production of white light and also the tuning of the visible emission as a function of the excitation wavelength.

Exploration into GeO_2_-based glasses containing Ag NCs has been limited, prompting the current investigation. These glasses possess noteworthy photonic-related characteristics, including a low cut-off phonon energy (between 500 and 800 cm^−1^) that is significant for reducing non-radiative losses, high refractive index when compared with silicate glasses, adequate properties for ultrafast devices, and a transmission region ranging from 400 to 5000 nm. Applications for these glasses have been reported, such as frequency upconversion [35], RE photoluminescence intensity enhancement with and without NPs [36], white light generation [35], pedestal waveguides for optical amplifier applications, and recently, nuclear shielding applications. The viscosity of these materials is influenced by the temperature used to anneal them, and the production of Ag NCs is facilitated in environments whose viscosity is higher. This aforementioned viscosity can be achieved by subjecting the material to annealing temperatures below the glass transition temperature, as demonstrated in previous reports [14,31]. Motivated by these findings, we have devised a method for cultivating Ag NCs within GeO_2_-PbO glasses by employing temperatures of 400 °C to anneal the material, which are situated below the temperature of the glass transition. Additionally, our research has revealed that elevated annealing temperatures, specifically at 470 °C, diminish the material’s viscosity, promoting the aggregation of Ag NCs and consequently facilitating the formation of Ag NPs [31]. The present study delves into the optical properties of GeO_2_-PbO glasses doped with Tm^3+^ ions containing Ag NCs. The investigation encompasses an analysis of absorbance, PL, and transmission electron microscopy (TEM) results. Furthermore, the study provides insights into the photoluminescence lifetime decay, elucidating the behavior of Ag NCs in the VIS region and Tm^3+^ ions in both the VIS and NIR regions, thereby shedding light on the ET mechanisms between these species. The research also explores the phenomenon of visible tunable light emission of Ag NCs under the influence of Tm^3+^ ions, which themselves emit in the visible range. The ET mechanisms among Ag NCs and Tm^3+^ ions have been previously documented in oxyfluoride glass systems. It is important to highlight that lead–germanate glasses exhibit distinct behavior compared to oxyfluoride glasses in the formation of silver nanoclusters. In oxyfluoride glasses, color centers form around F^−^ charges in the system [4,5,6]. In contrast, in lead–germanate glasses, these color centers exclusively originate around non-bridging oxygen sites through matrix-assisted reduction, which is the sole source of available negative charges [37].

The current investigation highlights the properties of GeO_2_-PbO glasses to serve as hosts for both Tm^3+^ ions and Ag NCs, while also elucidating the intricate ET processes among these entities. This work is also motivated by recent results of Yb^3+^-doped GeO_2_-PbO glasses that corroborated the ET mechanism between these species [32]. It also showed the possibility of hosting Ag NCs and Yb^3+^ ions in GeO_2_-PbO glasses. Upon optical excitation in the UV-blue region, the present study observed PL emissions in both the VIS and NIR regions, attributable to the combined emissions of Ag NCs and Tm^3+^ ions. These findings underscore the potential of such systems for applications in the development of exceptionally broad-band light sources, lasers, and solar cell coatings, among others.

## 2. Materials and Methods

Glass samples were composed of 40 wt.% GeO_2_ and 60 wt.% PbO (GP), with additions of Tm_2_O_3_ (1.5 wt.%) and AgNO_3_ (2.25/4.5 wt.%). Comparative samples containing only AgNO_3_ (4.5 wt.%) were also prepared. The glasses were labeled as: GP Tm, GP Tm 2.25% Ag, GP Tm 4.5% Ag, and GP 4.5% Ag. An alumina crucible was used to melt the reagents at 1200 °C for 1 h with mechanical stirring, which were swiftly cooled in ambient air using a preheated brass mold. Annealing at 400 °C (below the glass transition temperature) was performed for 1 h to reduce internal stresses. The annealed sample remained in the furnace, gradually cooling at a rate of approximately 1 °C per minute until it reached room temperature. This deliberate choice of annealing below the glass transition temperature was made to minimize the nucleation rate of Ag atoms dispersed within the glass matrix to favor Ag NC formation, as reported before [14,31].

Following this, the samples underwent precise polishing and cutting for characterization purposes. Absorption spectra were recorded using an OceanOptics QE65 PRO spectrometer, covering wavelengths from 400 to 800 nm, and an OceanOptics NIRQuest512 spectrometer, spanning the range of 900–1700 nm. PL measurements in different wavelengths (360, 380, and 400 nm) were used as excitation to perform PL measurements with a Varian Cary Eclipse fluorescence spectrophotometer in combination with the previously mentioned QE65 PRO spectrometer and a 3D printed adapter. This approach allows the collection of a higher-fidelity signal and solves some of the Varian Cary Eclipse limitations due to damage for detection above 700 nm. The chromaticity diagram (CIE-1931) was obtained from the PL spectra and used to evaluate the different emissions in the visible range. Additionally, correlated color temperature (CCT) was performed using McCamy isotherm equations [38], represented by Equations (1) and (2), as follows:(1)$$n=\frac{\left(x-x_{e}\right)}{\left(y-y_{e}\right)}$$
(2)$$T=an^{3}+bn^{2}+cn+d$$

The color temperature (*T*) is calculated with a polynomial function with the *x* and *y* coordinates derived from the CIE diagram (obtained from PL spectra) and correlates the color of the emission to the blackbody radiation (*T_c_*). The resulting CCT is then employed to describe various shades of white, with reference to daylight. Lower temperatures correspond to colors closer to the red region of the visible spectrum, while higher temperatures are associated with the blue region.

To investigate the ET from Ag NCs to Tm^3+^ ions, we measured the PL decay lifetimes of Ag NCs using the fluorescence spectrophotometer. The PL decay lifetime for Ag NCs was obtained using a fit based on a double exponential decay function, as represented by Equation (3).
(3)$$I\left(t\right)=A_{1}\cdot exp\left(\frac{-t}{\tau_{fast}}\right)+A_{2}\cdot exp\left(\frac{-t}{\tau_{slow}}\right)$$

The photoluminescence (PL) intensity, denoted as “*I*” in this equation, is expressed in terms of constants *A*_1_ and *A*_2_, where “*t*” represents the lifetime. The PL decay lifetimes, represented as *τ_fast_* and *τ_slow_*, correspond to the decay processes associated with spin-allowed (singlet–singlet and triplet–triplet) and spin-forbidden (singlet–triplet and triplet–singlet) electronic transitions, respectively. PL measurements in the NIR region were conducted with a CW diode laser that operates at 405 nm. The PL signal was collected in a direction perpendicular to the incident excitation beam and analyzed by a Newport Cornerstone 260 monochromator equipped with a photomultiplier tube and a connected lock-in amplifier. To perform the PL decay lifetime analysis of Tm^3+^ ions at 800 nm, we used the previously mentioned setup with a CW diode laser operating at 405 nm, complemented by a Keysight DSO1024A oscilloscope. The PL decay lifetimes were obtained by using a fit based on a single exponential decay function. We highlight that the CW laser operating at 405 nm had to be used due to the limitations of the fluorescence spectrometer and QE65 PRO spectrometer setup to determine the photoluminescence decay curves and obtain the short (*τ_fast_*) and long (*τ_slow_*) decay times with enough resolution. For the characterization of the Ag NCs, we employed TEM operating at both 200 and 300 kV, along with electron diffraction measurements of the samples. TEM sample preparation involved a series of steps, including milling, mixing with distilled water, and partial decantation. The suspended particles located at the mid-height of the reservoir were isolated and subsequently deposited onto ultra-thin carbon film-coated copper grids for analysis. It is important to note that all measurements were conducted under ambient conditions at room temperature.

## 3. Results

The results of the GP glass absorption containing AgNO_3_ and Tm_2_O_3_ are depicted in Figure 1. Within these spectra, distinct absorption bands corresponding to electronic transitions of Tm^3+^ ions from the ground state, namely (^3^H_6_→^1^G_4_), (^3^H_6_→^3^F_2.3_), (^3^H_6_→^3^H_4_), (^3^H_6_→^3^H_5_), and (^3^H_6_→^3^F_4_), are readily discerned. Notably, the absence of a plasmon absorption band in these samples suggests that the Ag species predominantly exist in the form of NCs, which represents the early step in the evolution towards the growth of Ag NPs. It is important to highlight that Ag NCs are characterized by the absence of the typical plasmon absorption band normally associated with well-defined crystalline structures of Ag NPs. Nonetheless, it should be noted that the possibility of metallic NP formation cannot be entirely dismissed, although it potentially occurs at a lower concentration compared to that of the prevailing NCs.

A broad PL band and tunable visible light emission can be observed for the GP 4.5% Ag sample reported in a previous work [39], which is attributed to NCs of varying sizes. As the Ag NC emission peak wavelength is size-dependent, the PL in the blue region corresponds to the emission by smaller-sized Ag NCs. On the other hand, the PL of the green and red regions is associated with the emission of larger Ag NCs [6,7,15]. For this sample, under different excitations in the UV spectrum, a redshift is observed, from greenish yellow at smaller wavelengths to orange at larger ones. This phenomenon serves as evidence for the existence of a substantial distribution of NCs with varying dimensions within the sample, offering the potential for a tailored selection of PL by manipulating the excitation wavelength.

The PL characteristics of Tm^3+^ ions, when excited at 360 nm via the (^3^H_6_→^1^D_2_) transition [40], are illustrated in Figure 2. Notably, the emission spectra reveal a prominent presence in the blue region of the electromagnetic spectrum, as demonstrated in the CIE chromaticity diagram as inset in Figure 2, predominantly attributed to the (^1^D_2_→^3^F_4_) transition that corresponds to the large emission at 455 nm.

For the GP Tm 2.25% Ag sample, the PL emissions within the visible spectrum became evident upon excitation at 360, 380, and 400 nm, as presented in Figure 3a. Specifically, under excitation at 360 nm, the pronounced peak corresponding to Tm^3+^ ion emission at 455 nm was observed. Notably, this emission retained its preeminence in the spectrum, similar to the sample without AgNO_3_. Moreover, for excitations at 380 and 400 nm, we noticed a broad PL band within the range of 480–700 nm, due to the addition of Ag NCs, and a shift to the yellow region with a CCT around 3500 K, obtained through Equations (1) and (2), as presented in Figure 3b. Additionally, the emission at 800 nm corresponding to the ^3^H_4_→^3^H_6_ transition of Tm^3+^ ions was noticed, a phenomenon resulting from ET from larger-sized NCs, indicating their efficiency in mediating ET processes. Notably, these emissions solely originate from such transfers, as there is no significant absorption by Tm^3+^ ions at 380 and 400 nm [40].

Comparing the results of the GP Tm 2.25% Ag sample with those of the GP Tm 4.5% Ag sample (Figure 4a), we observe that the PL at 455 nm (^1^D_2_→^3^F_4_) decreased under excitation of 360 nm, as illustrated in Figure 4a. This suggests that, owing to the abundance of Ag NCs surrounding Tm^3+^ ions, a significant portion of the excitation at 360 nm was absorbed for the emission in the 480–700 nm range from Ag NCs, resulting in the decrease of the one at 455 nm and consequently resulting in a warm white emission with a CCT of 3730 K, as presented in Figure 4b. Simultaneously, the peak at 800 nm nearly matched the intensity of the one at 455 nm.

For excitations at 380 and 400 nm, situated outside the Tm^3+^ ions absorption band of 357.7 nm, due to the ^3^H_6_→^1^D_2_ transition [40], we notice a considerable intensity increase of Ag NCs PL (Figure 4a) and also an ET to Tm^3+^ ions at 800 nm (^3^H_4_→^3^H_6_ transition); moreover, a shift from warm white to yellow (CCT of 3300 K) took place, as shown in Figure 4b. This outcome corroborates data regarding the capacity of larger-sized NCs to promote ET to the lower energy levels of Tm^3+^. It is expected that the increased doping, leads to a higher population of Ag NCs that emit in the red spectral region, and consequently, ET to the ^3^H_4_ level will become more pronounced.

The influence of Ag NC doping is illustrated in Figure 5, which shows the chromaticity diagram (CIE-1931) for fixed excitation at 360 nm, for GP Tm, GP Tm 2.25% Ag, and GP Tm 4.5% Ag. The coordinates of the diagram were obtained by using the results of Figure 2, Figure 3a and Figure 4a for 360 nm excitation. When exposed to the same excitation wavelength at 360 nm, the GP Tm sample exhibited a strong blue emission. In contrast, the GP Tm 2.25% Ag sample shifted towards a lighter blue emission, whereas the GP Tm 4.5% Ag sample presented a redshift with a warm white emission at 3730 K. This alteration in PL was associated with the increase in the larger Ag NC concentration emitting in the red region. It is crucial to note that the emission observed at 360 nm excitation resulted from the combined contributions of both Tm^3+^ ions and Ag NCs emission. This occurred because, at this wavelength, there is simultaneous excitation of Ag NCs and Tm^3+^ ions (^3^H_6_→^1^D_2_ transition), as mentioned previously.

Regarding the emission at 800 nm (under 405 nm excitation) for samples with Tm^3+^ ions, we notice a PL intensity increase with AgNO_3_ concentration, as illustrated in Figure 6. This phenomenon indicates a larger ET in the GP Tm 4.5% Ag sample with respect to the GP Tm 2.25% Ag one. Then, the larger concentration of Ag NCs results in species with increased average size [15] that enable a more efficient ET to Tm^3+^ ions at 800 nm. Lastly, it is noteworthy that the AgNO_3_ concentration growth gave rise to Ag NCs that exhibited adequate sizes for excitation at 380 and 400 nm, leading to a more effective transfer energy to Tm^3+^ ions, as can also be seen when comparing Figure 3a and Figure 4a.

Studies have unveiled a clear correlation in the PL spectrum of Ag NCs where shorter-wavelength emissions are attributed to smaller-sized NCs, whereas longer wavelengths are associated with larger NCs. Consequently, Ag NCs emitting in the blue region exhibit the smallest dimensions, and a systematic increase in their size results in a pronounced redshift of the PL [7,14].

In Figure 7, we present a simplified energy diagram that elucidates the Tm^3+^ ion transitions upon excitation at 360 nm, as well as the Ag NCs excitation wavelengths at varying UV wavelengths. These insights are derived from the previously discussed PL findings presented in Figure 3 and Figure 4. The energy levels of Ag NCs depicted herein draw upon Velázquez’s model [7]. In this model, S_0_ represents the ground state, S_1_ the excited singlet state, and T_1_ the excited triplet state. Electronic transitions span various regions of the electromagnetic spectrum: blue (S_1_→S_0_), green–yellow (T_2_→S_0_), yellow–red (T_1_→S_0_), near-infrared (S_1_→T_1_), and far-infrared (T_2_→T_1_, omitted from illustration). Spin-allowed electronic transitions involve singlet–singlet (S_1_→S_0_) and triplet–triplet (T_2_→T_1_) states, characterized by relatively short decay lifetimes. In contrast, transitions encompassing singlet–triplet (S_1_→T_1_) and triplet–singlet (T_2_→S_0_ and T_1_→S_0_) states are classified as spin-forbidden, exhibiting non-spontaneous behavior and long decay lifetimes. Ag NCs PL in the UV-blue region is attributed to short decay times (*τ_fast_*) governed by spin-allowed transitions between energy states. Conversely, PL emissions of Ag NCs in the green, yellow, red, and IR regions correspond to long decay times (*τ_slow_*) that are associated with spin-forbidden transitions.

The PL decay curves of Ag NCs were measured for excitation at 380 nm, and the signal detection wavelength was set to 600 nm due to the broad emission profile of Ag NCs (Figure 3a and Figure 4a). The time decay curve was fitted using Equation (3) (Figure 8a), and the resulting short (*τ_fast_*) and long (*τ_slow_*) decay times are presented in Table 1.

Table 1 highlights a similar reduction in both short and long lifetimes with the introduction of Tm^3+^ ions. This observation corroborates the ET among the Ag NCs and the Tm^3+^ ions, consistent with the PL results discussed previously. As shown before, samples containing Tm^3+^ ions and Ag NCs exhibit emission at 800 nm when excited at wavelengths that do not correspond to the Tm^3+^ ion’s excitation range.

Figure 8b and Table 1 show the results for excitation at 400 nm and detection at 600 nm. In this case, the slight increase in short lifetimes in the GP Tm 4.5% Ag sample could possibly be due to the small dimension of Ag NCs. These smaller NCs may not efficiently absorb excitation at 400 nm, potentially resulting in the reception of energy from the larger-sized ones. Conversely, a reduction in Ag NC lifetime within the sample containing Tm^3+^ ions was observed for the long lifetime, associated with spin-forbidden transitions (T_2_→S_0_, T_1_→S_0_, and S_1_→T_1_). This decrease is a clear indicator of ET to the surrounding Tm^3+^ ions. This effect, observed for excitation at 380 nm, aligns with the phenomenon of ET to the ions, stemming from the larger-sized Ag NCs.

The results of Tm^3+^ ion decay lifetime measurements under excitation at a 405 nm diode laser are presented in Figure 9 and Table 2. Regarding the 800 nm emission (^3^H_4_→^3^H_6_ transition), as previously illustrated in Figure 6, a noticeable enhancement can be seen with the growth of AgNO_3_ concentration from 2.25 to 4.5 wt.%. This observed trend is further supported by the increase in the lifetime of Tm^3+^ ions, as evidenced in Table 2, as it changed from 220 μs to 239 μs, indicating a higher ET from the Ag NCs in the sample containing 4.5 wt.% AgNO_3_. A similar behavior was observed for Yb^3+^ ions due to Ag NCs in oxyfluoride glasses [8] and for Pr^3+^ in oxyfluoro tellurite glasses [41]. We attribute the mentioned ET mechanism to the decay processes from S_1_→T_1_ and T_1_→S_0_ levels of Ag NCs to the ^3^H_4_ level of Tm^3+^ ions serving as the primary path of ET in this system.

TEM analyses were carried out on the GP 4.5% Ag and GP Tm 4.5% Ag samples, as illustrated in Figure 10. We highlight that the TEM results for the GP 4.5% Ag sample, used as a reference in the present study, were already reported [39], and further analysis is provided in this work.

In the case of the GP Tm 4.5% Ag sample, the histogram presented in Figure 10c demonstrates that the highest density of Ag NCs was centered around 3.5–4.0 nm in size. The diffraction patterns were conspicuously absent, as evidenced by the inset in Figure 10c, validating the amorphous state of Ag NCs. It is worth noting that both histograms were performed based on several TEM images for each sample.

In the GP 4.5% Ag sample, there was also clear evidence of Ag NCs and a lack of diffraction patterns, as shown in the inset in Figure 10d. However, these Ag NCs exhibited a size distribution primarily concentrated between 2.5 and 3.0 nm, as delineated in the accompanying histogram. The notable Ag NC size enhancement of the previous sample, with respect to this one, indicates a potential involvement of Tm^3+^ ions in a mechanism that favors the nucleation of Ag species, as already reported for Eu^3+^-doped glasses [42]. It is noteworthy to emphasize that the presence of these amorphous Ag NCs does not preclude the possibility of Ag NPs forming within the glass matrix.

## 4. Conclusions

This study presents a novel technology for the fabrication of Ag NCs within GeO_2_-PbO glasses doped with Tm^3+^ ions. The glasses were prepared using the melt-quenching method, employing an annealing temperature below the glass transition point to facilitate the growth of Ag NCs. Characterization of these samples encompassed absorption, luminescence, lifetime, and transmission electron microscopy. The absorption spectroscopy results confirmed the successful integration of trivalent RE ions into the glass matrix. Transmission electron microscopy and electron diffraction techniques were employed to discern the presence of Ag NCs and their amorphous structure. The fact that the Ag NCs were amorphous was further confirmed through electron diffraction measurements. Visible light emission from the blue to the yellow region was observed for samples containing both Tm_2_O_3_ and Ag NCs; an increase in Ag doping led to a redshift in the PL spectra. This shift was attributed to the growth of larger Ag NCs and to the local concentration enhancement of emitting species, favoring emission in the red region. This investigation delved into ET mechanisms, which were scrutinized through luminescence and lifetime measurements. These mechanisms were linked to the proximity of energy levels among Ag NCs and RE ions. The experimental results corroborated ET from Ag NCs to Tm^3+^ ions, demonstrated by the increased emission of the ^3^H_4_→^3^H_6_ transition of Tm^3+^ ions at 800 nm (under excitation at 380 and 400 nm), mainly for the sample with a higher concentration of Ag NCs, which can be due to the presence of larger Ag NCs. These larger Ag NCs are more suitable for excitation at 380 nm and 400 nm, and they demonstrate higher efficiency for transferring energy to Tm^3+^ ions. Concurrently, the reduced long lifetimes of Ag NCs with the addition of Tm^3+^ ions at 600 nm (excitations at 380 and 400 nm) and the prolonged lifetimes of Tm^3+^ ions at 800 nm (excitation at 405 nm) corroborated the ET processes. Therefore, we attribute the ET of the present work to the decay processes from the S_1_→T_1_ and T_1_→S_0_ levels of Ag NCs to the ^3^H_4_ level of Tm^3+^ ions serving as the primary path of ET in this system. These findings unveil the potential of GeO_2_-PbO glasses as a promising host medium for Ag NCs, affirming their viability for photonic applications. The implications extend to diverse technological domains, including coatings for solar cells, broad-band visible light sources, and devices featuring adjustable light emission. Additionally, the potential for CW-tunable lasers operating in the visible region of the electromagnetic spectrum holds promise, paving the way for further exploration involving different RE ions. To the best of our knowledge, there are few studies that have investigated this mechanism in other hosts. Moreover, the GeO_2_-PbO glasses were not studied to demonstrate the possibility to host Ag NCs with Tm^3+^ ions or to show the ET mechanism between them. The present work represents a contribution for those who are interested in hosting Ag NCs in oxide glasses and fills a lack in the literature, as mainly fluorophosphate and oxyfluoride glasses have been exploited up to now.

## Figures and Tables

**Figure 1 micromachines-14-02078-f001:**
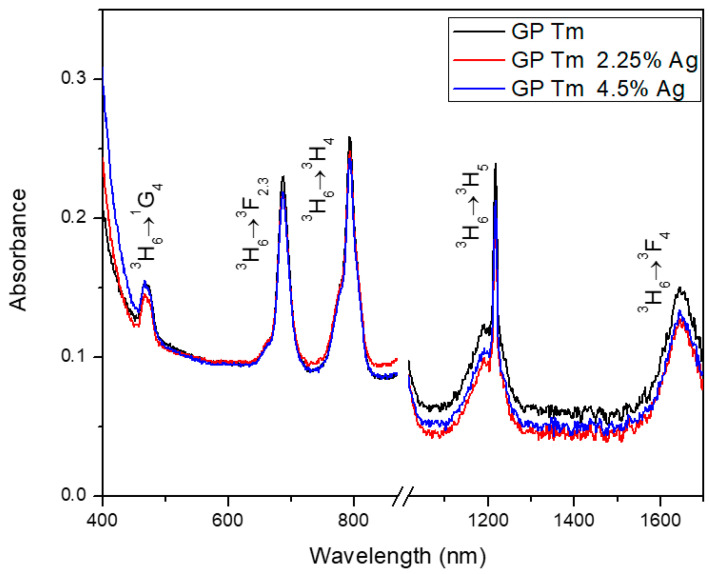
Absorption spectra of GP Tm, GP Tm 2.25% Ag, and GP Tm 4.5% Ag glass samples.

**Figure 2 micromachines-14-02078-f002:**
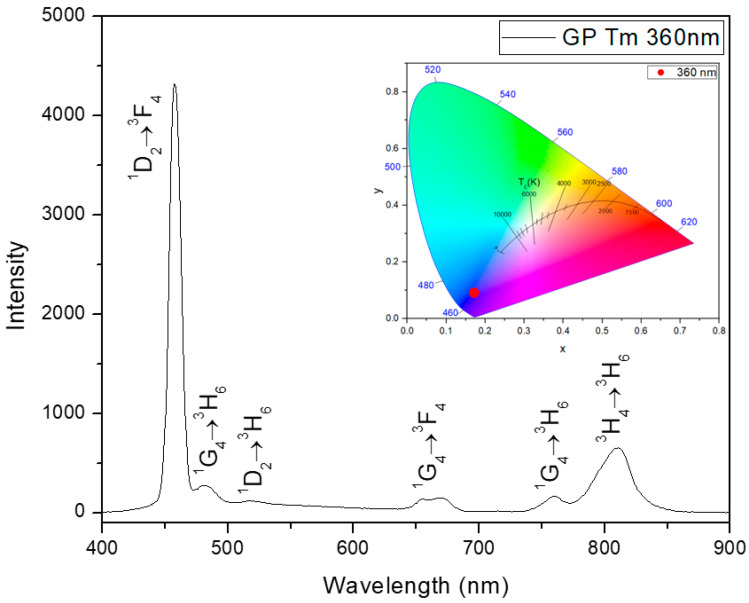
PL spectra of GP Tm sample at 360 nm excitation. The inset shows the sample chromaticity diagram (CIE-1931).

**Figure 3 micromachines-14-02078-f003:**
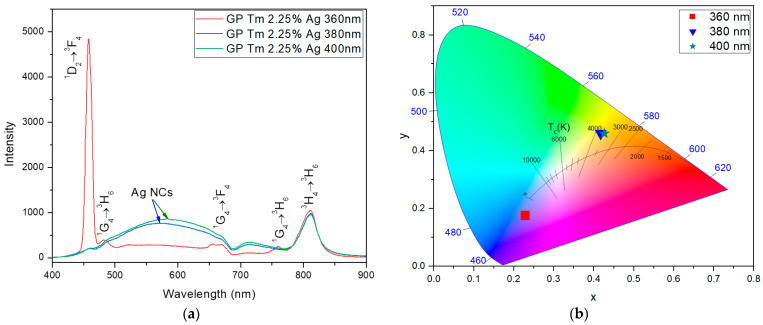
(**a**) PL spectra of GP Tm 2.25% Ag and (**b**) chromaticity diagram (CIE-1931) under varying excitation at 360, 380, and 400 nm.

**Figure 4 micromachines-14-02078-f004:**
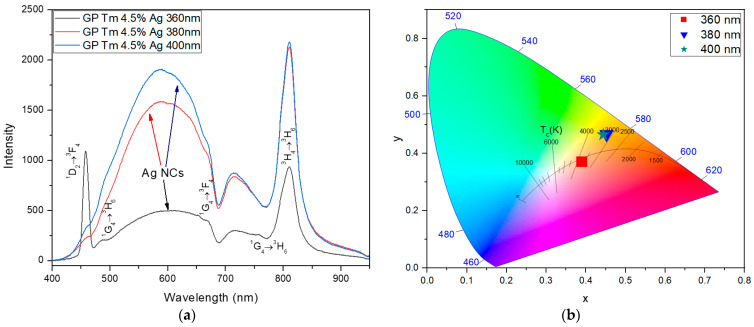
(**a**) PL spectra of GP Tm 4. 5% Ag and (**b**) chromaticity diagram (CIE-1931) under varying excitation at 360, 380, and 400 nm.

**Figure 5 micromachines-14-02078-f005:**
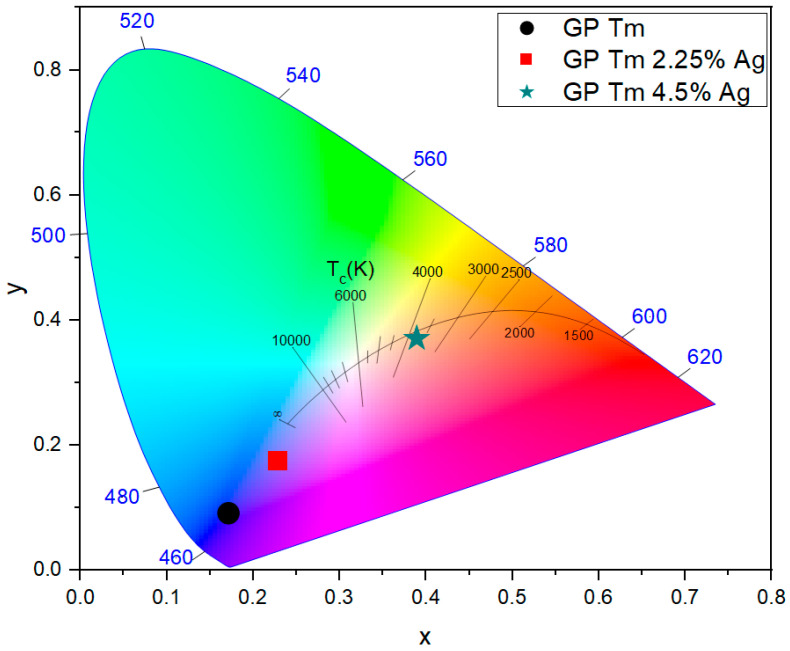
Chromaticity diagram (CIE-1931) for fixed excitation at 360 nm for GP Tm, GP Tm 2.25% Ag, and GP Tm 4.5% Ag.

**Figure 6 micromachines-14-02078-f006:**
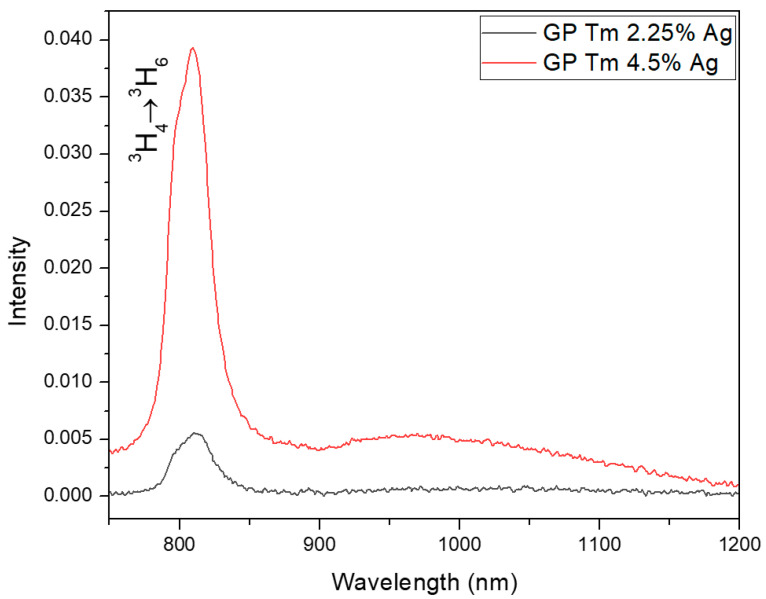
PL spectra of GP Tm 2.25% Ag and GP Tm 4.5% Ag samples under 405 nm CW laser diode excitation.

**Figure 7 micromachines-14-02078-f007:**
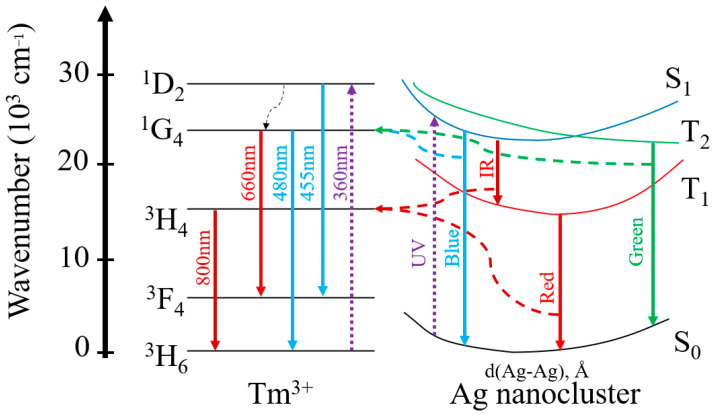
Simplified energy diagram of Ag NCs with ET to Tm^3+^ ions.

**Figure 8 micromachines-14-02078-f008:**
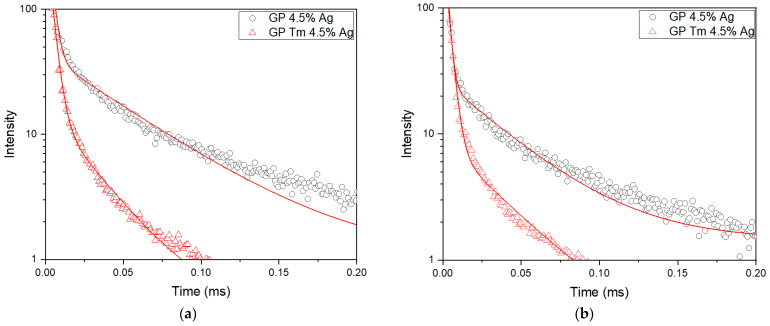
Ag NC PL decay curves of GP 4.5% Ag and GP Tm 4.5% Ag samples under (**a**) 380 and (**b**) 400 nm excitation and detection at 600 nm.

**Figure 9 micromachines-14-02078-f009:**
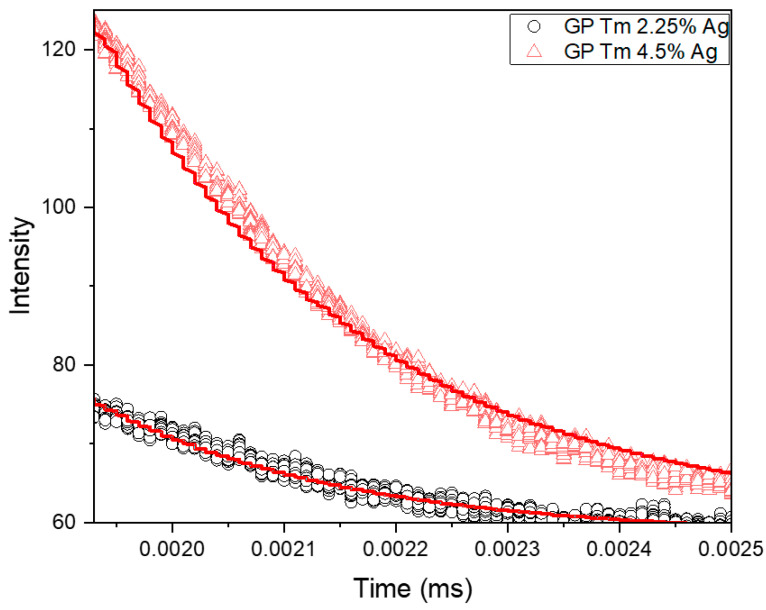
Tm^3+^ ions PL decay curves of GP Tm 2.25% Ag and GP Tm 4.5% Ag, with excitation at 405 nm and signal collected at 800 nm.

**Figure 10 micromachines-14-02078-f010:**
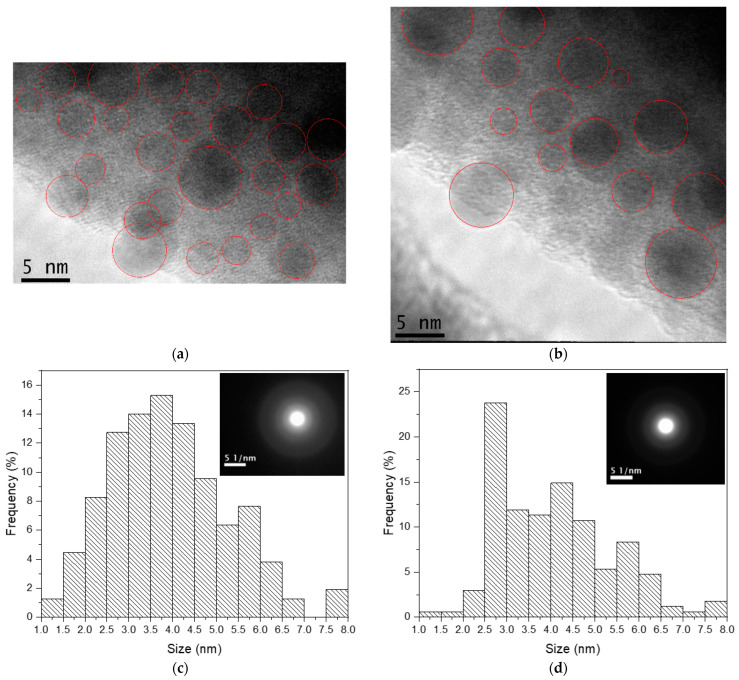
(**a**,**b**) TEM images and size distribution of (**c**) GP Tm 4.5% Ag and (**d**) GP 4.5% Ag with ED pattern as inset.

**Table 1 micromachines-14-02078-t001:** PL decay lifetimes (fast and slow components) of GP 4.5% Ag and GP Tm 4.5% Ag samples under 380 and 400 nm excitations and detection at 600 nm.

Sample	Excitation Wavelength (nm)	*τ_fast_* (μs)	*τ_slow_* (μs)
GP 4.5% Ag	380	11.7 ± 0.3	85 ± 2
	400	3.7 ± 0.4	72 ± 8
GP Tm 4.5% Ag	380	3.3 ± 0.1	26.2 ± 0.3
	400	5.8 ± 0.2	38 ± 2

**Table 2 micromachines-14-02078-t002:** PL decay lifetimes of Tm^3+^ ions for of GP Tm 2.25% Ag and GP Tm 4.5% Ag samples with excitation at 405 nm and detection at 800 nm.

Sample	Τime (μs)
GP Tm 2.25% Ag	220 ± 4
GP Tm 4.5% Ag	239 ± 9

## Data Availability

Data presented in this study are available upon request from the corresponding author.

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
