# Peer review of "Tunable Visible Light and Energy Transfer Mechanism in Tm3+ and Silver Nanoclusters within Co-Doped GeO2-PbO Glasses"

_micromachines, 2023, doi:10.3390/mi14112078_

Round 1
Reviewer 1 Report
Comments and Suggestions for Authors
This paper reports the formation of silver nanoclusters and nanoparticles in a lead germanate glass. The influence of silver on the spectroscopic properties of Tm3+ ions is adressed.
Synthesis and physical characterizations have been conducted in a rigorous way and conclusions are consistent with observations.
Reviewer 2 Report
Comments and Suggestions for Authors
The article discusses the luminescent properties of glasses of the original composition, GeO2-PbO co-doped by Tm3+ and silver nanoclusters. The authors found that silver nanoclusters are amorphous and their energy structure enhances the emission of Tm3+ ions in the longer wavelength range. The main conclusions of the article are confirmed by experimental studies (spectrophotometry and transmission electron microscopy) and their approximation. The article is presented competently, the material is presented in a logical sequence, however, there are minor inaccuracies in the experimental part. It is not entirely clear how photoluminescence was excited at 360 and 380 nm (Fig. 2, 3, 4) if CW diode laser operating at 405 nm was used (see lines 136 and 161)?
Reviewer 3 Report
Comments and Suggestions for Authors
The paper is good for the nolvelty, the experimental design and the results. So I recommond to publish this paper. But I have comments as follow. (1) For the Fig.9 and Table 2, the lifetimes 220 us and 239 us are almost the same. The lifetime change is not reasonable explaination for the energy transfer. Normally, the lifetime of accepter (Tm3+) is not changed with increasing the doping concentration of donor (Ag). (2) The spelling should be checked carefully.
Reviewer 4 Report
Comments and Suggestions for Authors
The authors Marcos Vinicius de Morais Nishimura et al. investigated the Tunable Visible Light and Energy Transfer Mechanism of GeO2-PbO Glasses doped with Tm3+ and co-doped Silver Nanoclusters.
I recommend this article to be published in your Journal after a major revision.
1. Some of the technical things, a period at the end of the sentence (row 131), and a space after the period at the end of the sentence (row 135), using different fonts (row 259), should be corrected throughout the manuscript.
2. XRD measurements should be done and included in the manuscript.
3. The authors should add Figure 5a where they would present the PL emission spectra of GP Tm, GP Tm 2.25% Ag and GP Tm 4.5% Ag recorded under the same conditioned excited at 360 nm.
4. Figure caption 5, the authors should add the excitation wavelength.
5. In the manuscript the authors didn’t explain the existence of hills in emission spectra. That isn’t marked with any transitions. Where do those hills come from? Are they from the matrix?
6. Did you try to vary the concentration of Tm3+?
7. Regarding the energy diagram, your emission spectra are recorded in the range of 400 nm, and excitation was 380 nm, explain in more detail what is the purpose of violet transition at 360 nm.
8. In TEM micrographs I can see that the particles that are not visible are circled. Can you do EDX or mapping to see the distribution of elements in the sample?
Comments on the Quality of English LanguageMinor editing of English language required.
Round 2
Reviewer 4 Report
Comments and Suggestions for Authors
The manuscript can be accepted for publication in its present form.